# An Intracellular Epitope of ASFV CD2v Protein Elicits Humoral and Cellular Immune Responses

**DOI:** 10.3390/ani13121967

**Published:** 2023-06-12

**Authors:** Wenlong Lu, Yilin Bai, Shuai Zhang, Xuyang Zhao, Jiaxin Jin, Xiaojing Zhu, Rui Wang, Yanan Wu, Angke Zhang, Gaiping Zhang, Guoqing Zhuang, Aijun Sun

**Affiliations:** 1College of Veterinary Medicine, Henan Agricultural University, Zhengzhou 450046, China; lwlhnnydx823@163.com (W.L.); onwardzs@163.com (S.Z.); zhaoxuyang17@163.com (X.Z.); jiaxinjin2020@163.com (J.J.); zhuxiaojing2020@163.com (X.Z.); ruiw_0623@163.com (R.W.); wlyananjiayou@yeah.net (Y.W.); zhangangke1112@126.com (A.Z.); zhanggaip@126.com (G.Z.); 2International Joint Research Center of National Animal Immunology, College of Veterinary Medicine, Henan Agricultural University, Zhengzhou 450046, China; 3Laboratory of Indigenous Cattle Germplasm Innovation, School of Agricultural Sciences, Zhengzhou University, Zhengzhou 450001, China; postbai@163.com; 4Longhu Laboratory of Advanced Immunology, Zhengzhou 450046, China

**Keywords:** African swine fever virus, CD2v, prokaryotic expression, monoclonal antibody, epitope, vaccine

## Abstract

**Simple Summary:**

African swine fever (ASF) causes acute hemorrhagic disease in domestic pigs. This disease causes huge economic losses worldwide, and there are no safe and effective vaccines. The ASF virus (ASFV) has at least 68 structural proteins. One of these is CD2v, an outer envelope protein usually used as a target for ASF vaccine development. However, the effective immunological regions of this protein have not been fully characterized. Herein, we used monoclonal antibodies to identify a novel B cell epitope located in the intracellular region of CD2v. Importantly, the epitope can elicit both humoral and cellular immune responses in a mouse model. This study provides new insights into ASF epitope vaccine development.

**Abstract:**

The African swine fever virus (ASFV) causes high mortality in domestic pigs. ASFV encodes an important protein target for subunit vaccine development, CD2v, but its most effective immunological regions are not known. Herein, we generated a monoclonal antibody (mAb) named IF3 by immunizing mice against the intracellular region of the CD2v protein (CD2v-IR). 1F3 specifically recognized CD2v, which is expressed transiently in transfected Sf9 cells and also in inactivated ASFV-infected porcine alveolar macrophage (PAM) cells. The epitope recognized by 1F3 is ^264^EPSPREP^270^, which is highly conserved in ASFV genotypes. Immunization of mice with this epitope elicited an increased IgG response, including IgG1 and IgG2a subtypes, and also increased CD8^+^ T cells and cytokine expression. Overall, these results indicate that this epitope induces both humoral and cellular immune responses that may be used for ASFV-related subunit vaccine design and development.

## 1. Introduction

African swine fever (ASF) is an extensive hemorrhagic disease caused by the African swine fever virus (ASFV). This virus mainly infects domestic pigs and wild boars, but it also infects soft ticks under some conditions [1,2]. Domestic pigs infected with highly virulent ASFV strains die 4–15 days post-infection, with a mortality rate of up to 100%. Those infected with moderately virulent strains show hemorrhagic symptoms with low mortality (30–70%). Finally, low virulence strains induce chronic infection without obvious vascular injury [3].

ASFVs have been classified into 24 genotypes based on the *B646L* gene [4]. In 2007, the ASFV genotype II strain spread into the Russian Federation and Eastern Europe through the Caucasus, and in 2017, it created an epidemic in Ukraine, Poland, and Irkutsk [5,6,7]. In August 2018, the first ASF was reported in Shenyang (Liaoning Province, China) and spread to Mongolia, Vietnam, Cambodia, Laos, North Korea, the Philippines, Myanmar, South Korea, and other Asian countries [8,9]. Although progress has been made in ASF vaccine development, no safe and effective vaccine is available [10,11]. The main protocol for ASF control includes quarantine and culling of infected pigs, which results in significant economic losses [2,12]. Therefore, there is an urgent need to develop safe and effective ASF vaccines.

ASFV has a 170–194 kb double-stranded DNA genome that contains over 150 open reading frames (ORFs) [1,13], encoding at least 68 structural proteins, most of which have unknown functions [13,14,15]. ASFV particles have a complex multi-envelope structure that includes the nucleoid, core shell, inner envelope, icosahedral capsid, and outer envelope [16]. Interestingly, one glycoprotein in the outer envelope of the virus, CD2v, encoded by the *EP402R* gene, is a homolog of the T cell adhesion molecule CD2 [15,17]. CD2v has a signal peptide, two extracellular immunoglobulin-like domains, a transmembrane (TM) region, and an intracellular region (referred to as CD2v-IR) that contains an acidic domain and proline-rich repetitive sequences [18]. This proline-rich region has been reported to interact with the mammalian actin-binding protein 1 (mAbp1), and this may control protein transport involved in immune regulation [19]. The CD2v-IR domain binds the trans-Golgi network (TGN) protein complex AP-1, affecting virus infectivity [20]. However, the precise role of the CD2v-IR domain in ASFV infection has not been determined.

The CD2v protein can activate both humoral and cellular immune responses in immunized animals [21,22]. Multiple linear B cell epitopes of CD2v have been identified recently, mostly in the extracellular region, but their function is unknown [22,23,24,25,26]. For example, we predicted epitope ^160^WNNSNINNFT^169^, which could effectively induce humoral and cellular immune responses in a mouse model [26]. In contrast, although some T cell epitopes have also been predicted in the CD2v-IR domain [22,27], epitope information in that region is scarce. Thus, we have generated a monoclonal antibody (mAb) named IF3 that specifically recognizes an epitope in the CD2v-IR domain, ^264^EPSPREP^270^. Its immunological effect was evaluated in a mouse model, and its potential use in ASFV-related subunit vaccine development is discussed.

## 2. Materials and Methods

### 2.1. Ethics Statements

The animal *study* was reviewed and approved by the ethics and animal welfare committee of Henan Agricultural University following the national guidelines for the care and use of laboratory animals (Approval number: SYXK-YU-2021-0003).

### 2.2. Plasmids, Cells, Serum, and Experimental Animals

The prokaryotic expression vector pET28b, human embryonic kidney 293T (HEK293T) cells, Sf9 cells, and hybridoma cells (SP2/0) were kept in our laboratory. *Escherichia coli* (*E. coli*) BL21 (DE3) competent cells were purchased from Takara Biomedical Technology (Beijing, China). ASF-positive serum was purchased from the China Veterinary Drug Supervision Institute (Beijing, China). SPF BALB/c mice were purchased from Beijing Vital River Laboratory Animal Technology Co., Ltd. (Beijing, China). Proteins from inactivated porcine alveolar macrophage (PAM) cells infected with ASFV were a kind gift from Professor Li Huang at the Harbin Veterinary Research Institute.

### 2.3. Prokaryotic Expression, Purification, and Identification of CD2v-IR Domain

The sequence of the *E402R* gene (ASFV Pig/HLJ/2018 strain, GenBank ID MK333180.1) was optimized and synthesized at Shanghai Sangon Bioengineering Co., Ltd. (Shanghai, China), inserted into a pET28b expression vector, and verified by restriction endonuclease digestion. Positive plasmids identified by PCR were sent to Shanghai Sangon Bioengineering Co., Ltd. (Shanghai, China) for sequencing. The recombinant plasmid pET28b-EP402R was designed to incorporate an *N*-terminal hexahistidine (6xHis) tag. The plasmid was transformed into BL21 (DE3) competent cells, which were inoculated into Luria-Bertani (LB) liquid culture medium containing kanamycin. Gene expression was induced with 0.2 mM isopropyl-β-D-thiogalactopyranoside (IPTG) at 25 °C for 16 h. BL21 (DE3) competent cells expressing the CD2v-IR domain were homogenized at low temperatures using an ultrahigh-pressure *E. coli* disrupter (Antox Nanotechnology, Suzhou, China). After centrifugation at 20,000× *g* for 60 min at 4 °C, the lysis supernatant was collected. The CD2v-IR domain was purified using a HisTrap FF (GE Healthcare, Anaheim, CA, USA) column and a HiLoad 16/600 Superdex 200 pg (GE Healthcare, CA, USA) column. Then, the protein was identified by Western blot, and purity was determined by sodium dodecyl sulfate-polyacrylamide gel electrophoresis (SDS-PAGE) and high-performance liquid chromatography (HPLC).

### 2.4. Eukaryotic Expression of Full Length CD2v and CD2v-IR Domain

Primers to amplify full-length CD2v and CD2v-IR from the *EP402R* gene (Appendix A) were designed using the synthesized gene as a template. The PCR product and pFastBacHTC vector were digested by *Bam*HI and *Hind*III and ligated overnight with T4 ligase at 16 °C. The recombinant plasmid was transformed into the TOP 10 competent cells. These were used to coat ampicillin-resistant LB solid culture media, from which positive recombinant bacteria were picked and plasmids were extracted. Recombinant plasmids were transformed into DH10Bac-competent cells and coated on LB solid culture media with kanamycin, tetracycline, gentamicin, IPTG, and X-Gal. Positive colonies were selected by screening for blue-white spots. Recombinant shuttle bacmid DNA was extracted and transfected into Sf9 cells to make recombinant baculovirus. After four passages, expression of both full-length CD2v and CD2v-IR was detected. The cell supernatant was collected and preserved at −80 °C.

### 2.5. Preparation of Monoclonal Antibody

Four-week-old female BALB/c mice were immunized with an 80 µg/mouse emulsified mixture of the CD2v-IR domain and Freund’s complete adjuvant. The mice were immunized again fourteen days later at a dose of 40 µg/mouse with the same emulsified mixture. The same protocol was followed for a third immunization 14 days after the second immunization. Ten days later, the mouse serum was isolated, and the antibody titer was determined using an indirect enzyme-linked immunosorbent assay (ELISA). The mice with the highest antibody titer were injected intraperitoneally with 100 µg non-adjuvant CD2v-IR domain to enhance immunity. The spleen B cells from the enhanced immunized mice were isolated and fused with SP2/0 cells. Positive hybridoma cells were screened by indirect ELISA. After two rounds of subclones, a single clonal hybridoma cell line was obtained. After extended culture, monoclonal antibodies were prepared and injected into the abdominal cavity of mice. The mouse monoclonal antibody allotype ELISA kit (Sino Biological Inc., Beijing, China) was used to determine the antibody subtypes according to the manufacturer’s instructions. Variable regions of the heavy and light chains of the monoclonal antibody were amplified by PCR with the primer sequences shown in Appendix A. Amplified products were recovered by a nucleic acid gel recovery kit (Vazyme, Nanjing, China), and sequences were analyzed at Shanghai Shenggong Biotechnology Co., Ltd. (Shanghai, China). Heavy chain (VH) and light chain (VL) regions of monoclonal antibodies were obtained.

### 2.6. Production of the CD2v-IR Fragments and Oligopeptides

The amino acid sequence of the CD2v-IR domain was analyzed by DNAstar Protean software (DNASTAR, Inc., Madison, WI, USA). According to the results of the antigenic index analyzed by DNAstar, CD2v-IR was first truncated into three fragments, C1 to C3. These fragments were amplified using the primers listed in Table 1, ligated to the pEGFP-Nl vector, and transfected into 293T cells. Then the N- and C-terminal amino acids of the C1 region were truncated into six fragments, C1-1 to C1-6, according to the Western blot results of the reaction between the fragments C1 to C3 and mAb 1F3. Gene fragments C1-1 to C1-6 were amplified using the primers listed in Table 1. These gene fragments were ligated to the pEGFP-Nl vector and transfected into 293T cells. According to the Western blot results of the reaction between these fragments and mAb 1F3, the gene fragments encoding polypeptides were synthesized at Shanghai Shenggong Biotechnology Co., Ltd. (Shanghai, China), cloned into the pEGFP-Nl vector, and expressed in 293T cells. To improve the access of the antibody to the antigen bound to the bottom of the ELISA plate or nitrocellulose membranes, peptides were synthesized by Kingsley Biotechnology Co., Ltd. (Nanjing, Jiangsu, China) and coupled to bovine serum albumin (BSA, Solarbio, Beijing, China) (Appendix A).

### 2.7. Western Blot

Western blot analysis was run to determine the reactivity of (i) purified His-tagged CD2v-IR domain against mAb or ASFV-positive serum, (ii) mAb 1F3 against CD2v protein expressed in Sf9 cells, and (iii) CD2v-IR truncations against mAb 1F3. The purified CD2v-IR domain, CD2v protein expressed in Sf9 cells, and truncations of the CD2v-IR domain were separated by electrophoresis on 12.5% SDS-PAGE gels and transferred to a polyvinylidene fluoride membrane (Sigma-Aldrich Trading Co., Ltd., Shanghai, China). After sealing at room temperature (RT) for 1 h with 5% dry milk, membranes were incubated with His-tagged mAb (1:5000) (Abcam, Cambridge, MA, USA), ASF positive serum (1:1000), or mAb 1F3 (1:500) for 2 h at RT. After washing with phosphate-buffered saline (PBS) with 0.01% Tween-20 (PBST) three times, membranes were incubated with HRP-conjugated goat anti-mouse IgG (ABclonal Technology Co., Ltd., Wuhan, China) or HRP-conjugated goat anti-swine IgG (Abcam, Cambridge, MA, USA) with a dilution of 1:3000 and 1:5000, respectively, at RT for 1 h. Hypersensitive chemiluminescence reagent (Enhanced Chemiluminescent, ECL) (New Cell & Molecular Biotech Co., Ltd., Suzhou, China) was added to the membranes, and Amersham Imager680 (GE Co., Ltd., Boston, MA, USA) was used to emit 1–5 min to identify the target protein.

### 2.8. Indirect Immunofluorescence Assay (IFA)

For the IFA assay, cell supernatants containing baculovirus were inoculated into 6-well plates containing cultured Sf9 cells that express full-length CD2v and CD2v-IR domains. After 72 h, the culture medium in the plate was discarded, and Sf9 cells were fixed with 4% paraformaldehyde. Triton X-100 (0.1%) (Solarbio, Beijing, China) was used for penetration. The plate was sealed with 10% FBS at RT and washed three times with PBS. The monoclonal antibody was diluted 1:100 with PBS, incubated at 37 °C for 1 h, and washed with PBS 3 times. Goat anti-mouse secondary antibody coupled with Alexa-Fluor-488 (Abcam, Cambridge, MA, USA) was diluted 1:500 with PBS, incubated for 1 h at 37 °C, and washed three times with PBS. Cellular DNA was stained with 4′,6-diamidino-2-phenylindole (DAPI, Beyotime, Shanghai, China). Cells were observed under a fluorescence microscope (Olympus Corporation, Tokyo, Japan) and an LSM 800 laser-scanning confocal microscope (Zeiss, Oberkochen, Germany).

### 2.9. Indirect ELISA and Dot Blot Assay

The CD2v-IR domain was diluted to 5 μg/mL with carbonate buffer solution (CBS), and 100 μL aliquots were added to each well in 96-well ELISA plates. The plates were incubated at RT for 2 h and sealed overnight with 300 μL of 5% dry milk at 4 °C. To each well, 100 μL aliquots of serum or hybridoma supernatants were added and incubated at 37 °C for 1 h. After washing with PBST, HRP-conjugated goat anti-mouse IgG (1:5000) was added to the plate and incubated at 37 °C for 30 min. The plate was incubated with 100 μL tetramethylbenzidine (TMB, Tiangen Biotech, Beijing, China) at RT for 5 min, and 50 μL of 2 M HCl was added to terminate the reaction. Results were expressed as optical density (OD) at 450 nm. ELISA and Dot blot assays were used to accurately map the epitope of mAb 1F3 to determine the smallest B cell epitope. In ELISA, 4 μg/mL BSA-coupled oligopeptides were coated on plates, and mAb (1:1000) or ASF-positive serum (1:1000) were used as primary antibodies. The other steps were the same as those performed for the hybridoma supernatants. In the dot blot assay, 10 μg of each BSA-coupled oligopeptide was spotted onto nitrocellulose membranes, and BSA was used as a negative control. After drying, membranes were blocked with 5% dry milk at 4 °C overnight and incubated with the mAb (1:500) or ASF-positive serum (1:500) at RT for 1 h. After washing three times with PBST, membranes were incubated with HRP-conjugated goat anti-mouse IgG (1:5000) and visualized in Amersham Imager680 (GE Co., Ltd., Boston, MA, USA) using ECL chemiluminescence reagent (New Cell & Molecular Biotech Co., Ltd., Suzhou, China).

### 2.10. Conservation Analysis of ASFV Strains

Epitopes were identified by mAb 1F3 in the Pig/HLJ/2018 strain and other 31 representative ASFV strains with 8 serotypes obtained from NCBI (https://www.ncbi.nlm.nih.gov/labs/virus/vssi/#/virus?SeqType_s=Nucleotide&VirusLineage_ss=African%20swine%20fever%20vrus,%20taxid:10497&utm_source=gquery&utm_medium=referral, accessed on 20 August 2022). The conservation of these epitopes was assessed by multiple sequence alignment using Unipro UGENE software v.42.0 (Unipro Center for Information Technologies, Novosibirsk, Russia).

### 2.11. Preparation of Peptide Vaccines

BSA (10 mg) was dissolved in 10 mL of 0.1 M PBS, and 3.3 mg of SULFO-SMCC (4-(*N*-Maleimidomethyl) cyclohexane-1-carboxylic acid 3-sulfo-*N*-hydroxysuccinimide ester sodium salt) was dissolved in 3 mL of N,N-Dimethylformamide (DMF). The SULFO-SMCC solution was slowly added to the BSA solution at RT, and the mixture was stirred for 1 h. The uncoupled SULFO-SMCC was removed with an ultrafiltration tube filled with 0.1 M PBS. The above conjugates were slowly added to the peptide solution (1 mg in 0.1 mL of 0.1 M PBS) and coupled in an ice bath with a magnetic stirrer for 12 h. After the reaction, the conjugates were ultrafiltered with 0.01 M PBS and characterized by SDS-PAGE gel electrophoresis.

### 2.12. Animal Immunization

A total of 20 mice were randomly divided into four groups: PBS negative control (0.1 mL), CD2v-IR (1 mg/mL, 0.1 mL), BSA (1 mg/mL, 0.1 mL), and polypeptide conjugated with BSA (1 mg/mL, 0.2 mL). The mice were immunized for the first time on day 0 and a second time on day 21. Immunization was performed by hind leg intramuscular injection. Tail blood samples were taken every seven days after immunization, and spleen samples were taken from mice two weeks after the second immunization (Figure 1).

### 2.13. Detection of Anti-CD2v IgG, IgG1, and IgG2a Antibodies

To perform the IgG, IgG1, and IgG2a antibody assay by ELISA, the serum was collected 7, 14, 28, and 35 days after immunization [28,29]. Briefly, 100 μL of the purified CD2v-IR domain was coated in 96-well plates (2 μg/mL) for 3 h at 37 °C. After blocking, coated plates were incubated with mouse serum in a sequential dilution of 1:200, 1:400, 1:800, 1:1600, 1:3200, 1:6400, 1:12,800, 1:25,600, 1:51,200, 1:102,400, and 1:204,800 for 1.5 h at 37 °C. Then, 100 μL of a 1:100,000 dilution of HRP-conjugated goat anti-mouse IgG and a 1:5000 dilution of IgG1 and IgG2a (Proteintech, Wuhan, China) were incubated for 60 min at 37 °C. After extensive washing, tetramethyl benzidine (TMB, Solarbio, Beijing, China) substrate was added for 15 min at 37 °C, and the reaction was stopped with 2 M H_2_SO_4_. The OD value was read at 450 nm by an automatic ELISA reader (BioTek, Winooski, VT, USA).

### 2.14. Detection of T Cell Subtypes

The mouse spleens were collected two weeks after the second immunization. Splenic lymphocytes were isolated with a lymphocyte separation solution (5 × 10^6^ cells/mL) and incubated with PE-labeled CD3, FITC-labeled CD4, and APC-A700-labeled CD8 (Proteintech, Wuhan, China) at 4 °C for 30 min. Fluorescence changes were monitored on a Beckman Coulter CytoFLEX S flow cytometer (Beckman Coulter, Inc., Brea, CA, USA) to evaluate the expression of CD3, CD4, and CD8.

### 2.15. Detection of Interleukin-4 (IL-4) and Interferon-γ (IFN-γ)

For the evaluation of cellular immunity, the production of IL-4 and IFN-γ in mouse serum from all experimental groups was measured two weeks after the second immunization using a commercial cytokine assay kit (Shanghai Enzyme Linked Biotechnology Co., Ltd., Shanghai, China), which was used according to the manufacturer’s instructions.

### 2.16. Statistical Analysis

Statistical data were analyzed using GraphPad Prism 9 software (GraphPad Software Inc., San Diego, CA, USA). Experimental data is presented as mean ± standard deviation (SD). Statistical analyses were performed using unpaired *t*-tests or two-way analysis of variance (ANOVA) F-statistics, and differences were considered significant at *p* < 0.05.

## 3. Results 

### 3.1. Expression and Purification of CD2v-IR Domain

To obtain the CD2v-IR domain, the truncated *E402R* gene fragment was ligated to the pET28b vector, and the pET28b-EP402R was transformed into BL21 (DE3)-competent cells. This domain was soluble in *E. coli,* and SDS-PAGE was consistent with its expected size (~17 kDa) (Figure 2A). The domain was first purified by cobalt ion column affinity chromatography using a His-tag antibody, followed by gel filtration chromatography. Purity was further confirmed by HPLC (Figure 2B) and SDS-PAGE (Figure 2C). Western blot analysis showed good antigenicity with both His-tag antibodies (Figure 2D) and ASF-positive serum (Figure 2E). However, a band appeared above 25 kDa (Figure 2D,E), possibly due to dimerization. These results indicate successful expression and purification of the CD2v-IR domain.

### 3.2. Production and Characterization of mAb against the CD2v-IR Domain

To prepare mAbs against the CD2v-IR domain, female BALB/c mice were immunized with purified CD2v-IR. After fusion with hybridoma cells, the monoclonal fusion cell line 1F3 was obtained after two rounds of subclone screening. Western blot analysis indicated that 1F3 could recognize both the full-length CD2v and CD2v-IR domains expressed in recombinant baculovirus (Figure 3A,B). The upper bands in Figure 3A may be caused by other proteins present after cell lysis that show non-specific interaction with the antibodies. 1F3 also recognized CD2v expressed in inactivated ASFV-infected PAM cells (Figure 3C). An IFA assay showed that 1F3 could bind both full-length CD2v and its CD2v-IR domain expressed in Sf9 cells (Figure 3D). Both protein species were found expressed in the cytoplasm of Sf9 cells, indicating that mAb 1F3 can specifically recognize both full-length CD2v and CD2v-IR domains (Figure 3E). Homotype determination indicated that 1F3 is an IgG1 subclass (Table 2). IF3 nucleotide sequences of the VH and VL variable regions were amplified by PCR (Table 3). Overall, these results indicate the successful generation of an mAb that recognizes the CD2v-IR domain.

### 3.3. Identification of a Highly Conserved Linear B Cell Epitope, ^264^EPSPREP^270^


To identify the epitopes recognized by 1F3, three overlapping peptides (C1 to C3) of the CD2v-IR domain (230–360 residues) were expressed in 293T cells. Western blot results showed that the epitopes recognized by 1F3 were located in the Cl region (residues 230–290). This region was further truncated *N*- and C-terminally to produce six shorter fragments (C1–1 to C1–6) that were expressed in 293T cells. 1F3 recognized an epitope localized in the overlap between C1–4 (262–336) and C1–6 (230–275aa), i.e., residues 262–275 (Figure 4B). The corresponding gene fragments were synthesized (Appendix A) and expressed in 293T cells. Western blot analysis indicated that the bona fide epitope region is ^264^EPSPREP^270^ (Figure 4C). We found that this epitope is located at an antigenic index peak of segment C-1 (Figure 4D). Thus, we synthesized the ^264^EPSPREP^270^ peptide and coupled it to BSA (Appendix A) to determine the minimum binding site recognized by 1F3. The smallest linear epitope recognized by 1F3 was ^264^EPSPREP^270^, confirmed by ELISA (Figure 5A) and dot blot (Figure 5B) assays, which also showed that ^264^EPSPREP^270^ reacts with ASF-positive serum (Figure 5C,D). These results suggest that the epitope ^264^EPSPREP^270^ acts as an immunogen to elicit immunological responses. Comparison with 32 ASFV strains shows that this epitope is highly conserved in ASFV genotypes (Appendix A).

### 3.4. The Epitope ^264^EPSPREP^270^ Induced High Antibody Titer in Humoral Immune Response

To estimate the potential role of the epitope ^264^EPSPREP^270^ on the humoral immune response, we first evaluated the IgG isotope profiles in serum from mice immunized with the CD2v-IR domain (Figure 1). As expected, ELISA results showed that IgG levels increased gradually 35 days after immunization (Figure 6A). Both IgG1 and IgG2a levels increased after immunization (Figure 6B,C). ELISA results showed that epitope ^264^EPSPREP^270^ increased IgG, IgG1, and IgG2a after BSA-peptide immunization relative to the BSA control (Figure 6D–F). These results revealed that the epitope ^264^EPSPREP^270^ can successfully elicit a humoral immune response in a mouse model.

### 3.5. Effect of Epitope ^264^EPSPREP^270^ on Cellular Immune Response

To investigate whether the epitope ^264^EPSPREP^270^ affects the cellular immune response, lymphocytes were isolated from mouse spleens, and T-cell subsets were determined by flow cytometry. To determine the levels of IFN-γ and IL-4 by ELISA (Figure 1), serum was collected from immunized mice. Compared to the control group, CD2v-IR domain immunization increased expression of IFN-γ and IL-4 (Figure 7A) and CD4^−^CD8^+^, but not CD4^+^CD8^−^, T lymphocytes (Figure 7B). Immunization by epitope ^264^EPSPREP^270^ elicited significant expression of IFN-γ and IL-4 compared to the control group (Figure 7C) and CD4^−^CD8^+^ T lymphocytes, but reduced CD4^+^CD8^−^ T lymphocytes (Figure 7D). These results indicate that the epitope ^264^EPSPREP^270^ specifically elevates CD4^−^CD8^+^ cellular immune responses.

## 4. Discussion

The CD2v protein is encoded by the *EP402R* gene and is the major membrane protein on the surface of ASFV viral particles [18]. Deletion of this gene completely attenuated the ASFV BA71 strain (BA71ΔCD2v), which suggests that immunization of domestic pigs with this deletion mutant could be used against both the homologous BA71 virus and the heterologous E75 virus [30]. Intranasal vaccination with the deletion mutant conferred dose-dependent cross-protection against lethal ASFV infection [31], but this deletion did not reduce virulence in domestic pigs infected with the ASFV Georgia strain [32]. Reduced virulence was observed with the double gene deletion mutant ASFV-SY18-ΔCD2v/UK, which interestingly triggered a protective immune response and showed protection against the lethal parental virus [33]. Additionally, CD2v and C-type lectin protein (EP153R) mediate the serological specificity of haemadsorption inhibition (HAI), which is important in preventing homologous ASFV infection [34,35]. Overall, these findings suggest that CD2v plays a distinctive role in the pathogenesis of different ASFV strains, but using an unknown molecular mechanism.

As an immunogen, CD2v can elicit adaptive immune responses [22,36]. Indeed, modified vaccinia virus Ankara (MVA) expressing CD2v, P72, and EP153R induced significant cellular immunity characterized by T cell proliferation and IFN-γ production [37]. For rBartha-K61-pASFV expressing CD2v, P72, P30, pp62, and P54, high production of antibodies was observed in piglets and mice [38]. Importantly, a baculovirus-based expression vector expressing the extracellular domains of CD2v, P54, and P30 induced specific T-cell immunity directly after in vivo immunization and partially protected pigs against a sublethal attack with ASFV [39]. However, due to the absence of key immunogens, there are still no safe and effective subunit vaccines against ASFV [40]. Epitope vaccines constitute an alternative strategy for subunit vaccine development. In severe acute respiratory syndrome coronavirus 2 (SARS-CoV-2) infection, CoVac-1 is a T cell epitope-based vaccine candidate that showed a favorable safety profile and induced strong T cell immune responses [41]. Another vaccine candidate, rLaSota/SBNT, containing four neutralizing epitope domains and four T-cell epitope peptides, induced heterologous infectious bronchitis virus (IBV)-specific neutralizing antibodies and T-cell responses. rLaSota/SBNT provided significant protection against challenge by homologous IBVs [42]. A polytope DNA vaccine (pVAX1-rTEM) consisting of B cell and T cell epitopes from the Tembusu virus (TMUV) envelope € protein elicited both humoral and cellular immune responses in ducks against a TMUV challenge [43]. Polydopamine (PDA) nanoparticles conjugated with the BPP-V and BP-IV epitope peptides provided effective protection against avian influenza virus (AIV) viral infection [28]. ASFV T cell epitope peptides were identified by the ELISpot assay, providing the basis for the development of epitope vaccines [44,45]. In CD2v-IR, two predicted T cell epitopes were identified by the IFN-γ ELISpot assay, eliciting a strong T cell response, which may benefit epitope-based vaccine development [22].

Multiple ASFV linear B cell epitopes have been identified in the CD2v extracellular domain by mAbs. For example, the three linear B cell epitopes ^147^FVKYT^151^, ^157^EYNWN^161^, and ^195^SSNY^198^ were recognized by five mAbs generated by baculovirus-expressed truncated CD2v protein [23]. A linear epitope formed by amino acids 28–51 was identified by mAbs generated by truncated CD2v protein fused with Norovirus (NoV) P particles [25]. Two linear B cell epitopes, ^128^TCKKNNGTNT^137^ and ^148^VKYTNESILE^157^, were recognized by mAbs made by CHO-S-expressing truncated CD2v [24]. The ^38^DINGVSWN^45^ and ^134^GTNTNIY^140^ epitopes were identified by peptide scanning using mAbs generated by eukaryotic cells expressing CD2v [46]. It is noteworthy that some of the linear B cell epitopes mentioned above overlap; therefore, the immunological activity of these epitopes needs to be evaluated. We recently predicted and identified the dominant linear B cell epitope ^160^WNNSNINNFT^169^, which induces humoral and cellular immune responses in a mouse model, strongly suggesting that the linear B cell epitope may facilitate ASFV subunit vaccine design and development [26]. Here, we identify a novel highly conserved epitope, ^264^EPSPREP^270^, located in the CD2v-IR domain that can be used for subunit vaccine design and development.

For immunological activity evaluation, we separately immunized mice with the same amount of synthesized epitope-peptide-conjugated BSA protein and CD2v-IR domain (Figure 1). As expected, the CD2v-IR domain induced both humoral and cellular responses (Figure 6A–C) [47,48], whereas the epitope ^264^EPSPREP^270^ induced higher humoral activity than BSA, with increased IgG antibodies (Figure 6D) and subtypes IgG1 (Figure 6E) and IgG2a (Figure 6F). Surprisingly, this epitope also induced high expression of IL-4 and IFN-γ (Figure 7C). IgG1 and IL-4 are related to humoral immune responses, whereas IgG2a and IFN-γ are associated with cellular immune responses [28,29]. Our results indicate that this epitope can efficiently elicit both humoral and cellular responses. In stimulated CD8^+^ T cells, a significantly higher increase was also observed in this epitope group than in the BSA group (Figure 7D), but after epitope stimulation, CD4^+^ T cells were lower (Figure 7D). Another epitope located in the extracellular region of CD2v has been reported to activate specific CD4^+^ T cells in a mouse model [26]. We speculate that the cellular immune response induced by epitope ^264^EPSPREP^270^, promoting CD8^+^ T cell metabolism but inhibiting CD4^+^ T cell metabolism, may contribute to the efficient clearance of viral infection [49,50]. 

Although we screened a B-cell linear epitope in the CD2v-IR domain and tested its immunogenicity, the immunity of a single epitope is limited. Therefore, more research is needed on the immunodominant epitope in the CD2v-IR domain.

## 5. Conclusions

We generated and characterized a mAb and identified an immunodominant epitope in CD2v protein, ^264^EPSPREP^270^, which is highly conserved in different ASFV genotypes. This epitope can elicit both humoral and cellular immune responses. Our findings provide a solid foundation for the further study of the antigenic function of ASFV CD2v and for the development of diagnostic methods and effective ASFV-related subunit vaccines. 

## Figures and Tables

**Figure 1 animals-13-01967-f001:**
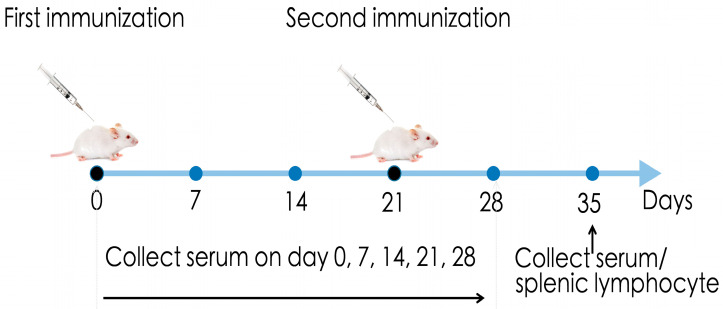
Experimental procedures of the immunization and sample collection. Mice were immunized on days 0 and 21 by leg intramuscular injection, serum was collected on days 0, 7, 14, 21, 28, and 35 post immunization, and splenic lymphocytes were collected on day 35 post immunization.

**Figure 2 animals-13-01967-f002:**
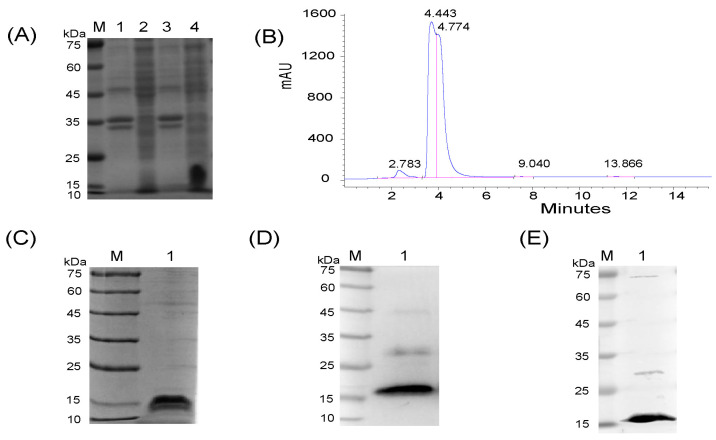
Expression and identification of the CD2v-IR domain. (**A**) SDS-PAGE analysis; M: protein marker; lanes 1–2: precipitates (lane 1) and supernatant (lane 2) of uninduced lysed cells; lanes 3–4: precipitates (lane 3) and supernatant (lane 4) of induced lysed cells; (**B**) HPLC analysis; (**C**) identification by SDS-PAGE; (**D**,**E**) Western blot with His-tag antibody (**D**) and ASFV-positive serum (**E**). In panels (**C**–**E**), M is the protein marker, and lane 1 is the target protein.

**Figure 3 animals-13-01967-f003:**
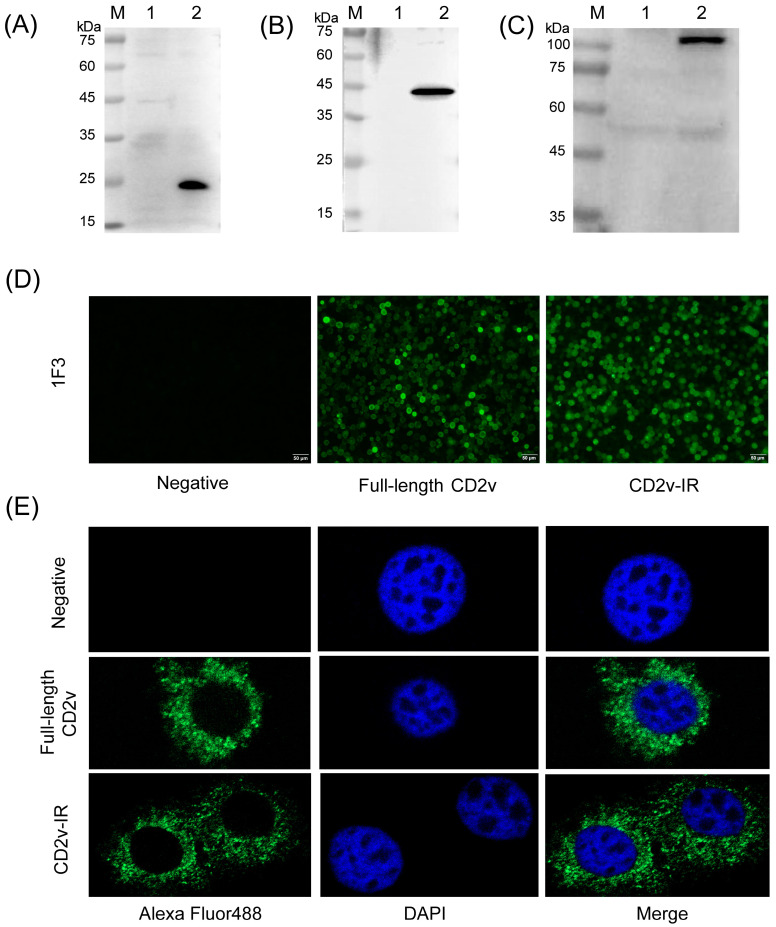
Characterization of mAb 1F3. (**A**,**B**) Western blot analysis using 1F3 of CD2v-IR domain (**A**) or full-length CD2v protein (**B**) expressed in Sf9 cells: M, protein marker; lane 1: Sf9 cells; lane 2: target protein expressed in Sf9 cells; (**C**) Western blot analysis using 1F3 of CD2v expression in inactivated ASFV-infected PAM cells: M, protein marker; lane 1: PAM cells; lane 2: inactivated ASFV-infected PAM cells; (**D**) specificity analysis of 1F3 by IFA assay; (**E**) localization of CD2v in Sf9 cells.

**Figure 4 animals-13-01967-f004:**
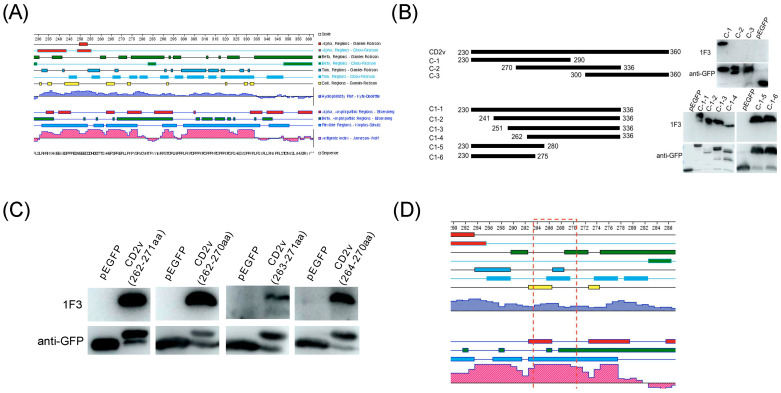
Mapping the epitope by mAb 1F3. (**A**) Antigenic index of CD2v-IR domain analyzed by DNAstar software; (**B**) Western blot analysis of truncations of peptide CD2v (230–360); (**C**) Western blot analysis of synthesized peptides; (**D**) epitope analysis by DNAstar software.

**Figure 5 animals-13-01967-f005:**
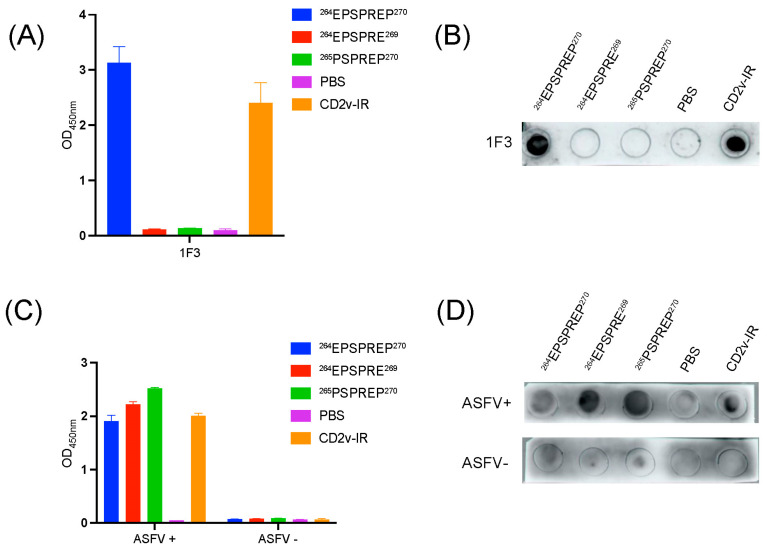
Epitope analysis by indirect ELISA and dot blot. (**A**,**B**) ELISA (**A**) and dot blot (**B**) analysis of the epitopes using 1F3. (**C**,**D**) Same as A–B using ASF positive serum. ASFV+: ASF positive serum; ASFV−: ASF negative serum.

**Figure 6 animals-13-01967-f006:**
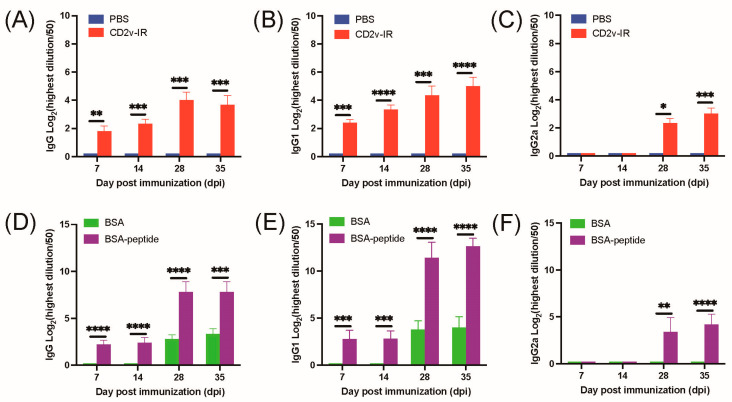
Effect of the CD2v-IR domain and epitope peptide on antibody production. Immunized mouse serum was collected on days 7, 14, 28, and 35 post immunization. (**A**–**C**) Induction of IgG (**A**), IgG1 (**B**), and IgG2a (**C**) by the CD2v-IR domain; (**D**–**F**) Same as (**A**–**C**) for the epitope peptide. Titers were analyzed by ELISA. The PBS and BSA groups were used as controls. *, *p* < 0.05; **, *p* < 0.01; ***, *p* < 0.001; ****, *p* < 0.0001.

**Figure 7 animals-13-01967-f007:**
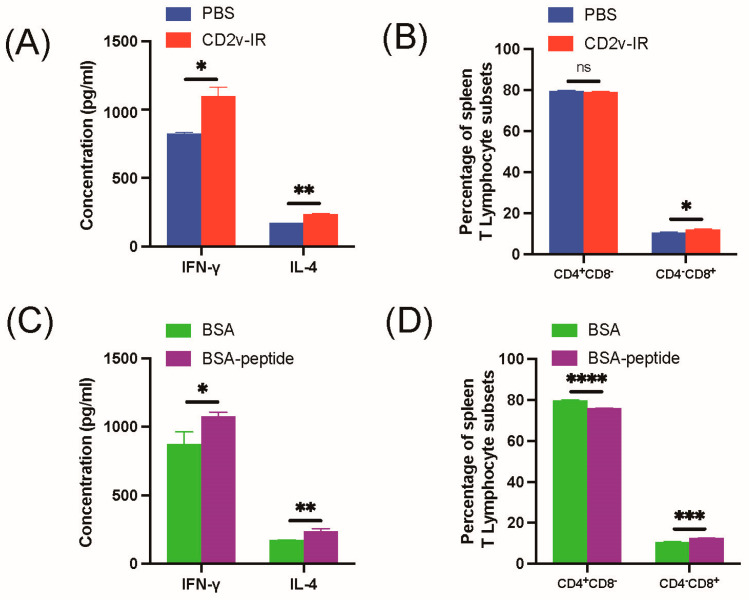
Effect of the CD2v-IR domain and epitope peptide immunization on the cellular immune response. (**A**,**B**) IL-4 and IFN-γ detection by ELISA in mouse serum collected on day 35 after the second immunization with CD2v-IR domain (**A**) and percentage of CD4 and CD8 T lymphocytes detected by flow cytometry; (**C**,**D**) same as A-B for epitope peptide immunization. *, *p* < 0.05; **, *p* < 0.01; ***, *p* < 0.001; ****, *p* < 0.0001; ns represent non-significant, *p* > 0.05.

**Table 1 animals-13-01967-t001:** Primers for the truncated protein CD2v.

Primers	Sequences (5′–3′)
pEGFP-N1-230-290-F	gctagcATGTCCCTCCGCAAGCGCAAGAA
pEGFP-N1-230-290-R	ggatccGTGCGCATGTAGTAGATGGGGGT
pEGFP-N1-270-336-F	gctagcATGCCTCTGCTGCCTAAGCCCTA
pEGFP-N1-270-336-R	ggatccGTGGAAGGGAGAGGCTTGGGAG
pEGFP-N1-300-360-F	gctagcATGCTCCCCAAGCCTTGCCCTCCTCCTAAGCCTTGCCCCCCTCCCAAGCCCT
pEGFP-N1-300-360-R	ggatccGTGATGATGCGATCGACGTGGATGA
pEGFP-N1-241-336-F	ctcgagATGATCGAGTCCCCTCCTC
pEGFP-N1-251-336-F	ctcgagATGGAGGAGCAATGCCAAC
pEGFP-N1-262-336-F	ctcgagATGATCCACGAGCCCTCCCCT
pEGFP-N1-230-280-R	gatcggatccGTCTGGTAGCGAGAGTAGGGCTT
pEGFP-N1-230-275-R	gatcggatccGTGGGCTTAGGCAGCAGAGGCT

**Table 2 animals-13-01967-t002:** Isotype test of CD2v mAb.

Clone	1F3
IgG1	1.5
IgG2a	0.03
IgG2b	0.03
IgG2c	0.02
IgG3	0.04
Blank	0.02

**Table 3 animals-13-01967-t003:** Variable region of the heavy and light chain of CD2v mAb.

mAb	Heavy Chain	Light Chain
1F3	QLQESGGGLVQPGGSMKLSCAASGFPFSDAWMDWVRQSPERGLEWVAEIRSKANNHATYYAESVKGRFTISRDDSKSRVYLHMNSLRGEDTGIYYCNGPAAGVWGQGTTVTVSS	DIELTQSPVSITASRGEKVTITCRANSSISSNNFHWYLRRPGSSPKLLIYRTSILASGVLDSFSGRGSESSSTLTIHYMQDEVAATYYCQQGSICPPRSEGGPSSRSN

## Data Availability

All data presented in this study are included within the article and its Appendix A.

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
