# Peer review of "An Intracellular Epitope of ASFV CD2v Protein Elicits Humoral and Cellular Immune Responses"

_animals, 2023, doi:10.3390/ani13121967_

Round 1

Reviewer 1 Report

Line 47 – sentence structure is confusing. Subject verb agreement is incorrect. This sentence needs to be totally rewritten.

Line 67 – “elusive” is not an appropriate word choice. Suggestion: the precise role of 66 CD2v-IR in ASFV infection has not been determined.

Line 148 – Why were these specific peptides selected? The logic for selecting the primers for peptide synthesis needs to be explained further and justified. This selection process is key to the work.

Line 150 – What proteins and why do you need the Western blots?  This will improve readability.

Line 202 – a flow chart of peptide vaccine production would be useful. This is actually a relatively complex approach to vaccine production – even for experimental purposes.

Line 215 – please explain this phrase more clearly: and nonimmune as a blank control 215 group.

Line 220 – figure #’s are out of order – this is confusing.

Line 224 – reference?

Line 247 – Justify the statistical analysis approach. Are the data expected to be normally distributed, were they so distributed?

Line 282 – I am not sure the description of the technical work describing generation of the mAb is essential for the paper. A brief description of the specificity would be enough.

Line 315 – This needs to be very clearly delineated as swine serum?  This may be the strongest validation of this approach in the paper.

Line 318 – What is a “level”? Do you mean titer, antibody activity, etc.?

General Comments:

The paper is well written and with attention to some details as mentioned can be a very good manuscript. However, it lacks one fundamentally important bit of data: Are these epitopes immunogenic in swine and would or do they generate a potentially protective response in swine. ASFV is a porcine pathogen, and all immunological data needs to be validated in context of host species responses.

Depending on how the authors respond, this can be a very good paper. The mouse immunology needs greater validation for relevance to a swine pathogen.

Line 47 – sentence structure is confusing. Subject verb agreement is incorrect. This sentence needs to be totally rewritten.

Line 67 – “elusive” is not an appropriate word choice. Suggestion: the precise role of 66 CD2v-IR in ASFV infection has not been determined.

Line 148 – Why were these specific peptides selected? The logic for selecting the primers for peptide synthesis needs to be explained further and justified. This selection process is key to the work.

Line 150 – What proteins and why do you need the Western blots?  This will improve readability.

Line 202 – a flow chart of peptide vaccine production would be useful. This is actually a relatively complex approach to vaccine production – even for experimental purposes.

Line 215 – please explain this phrase more clearly: and nonimmune as a blank control 215 group.

Line 220 – figure #’s are out of order – this is confusing.

Line 224 – reference?

Line 247 – Justify the statistical analysis approach. Are the data expected to be normally distributed, were they so distributed?

Line 282 – I am not sure the description of the technical work describing generation of the mAb is essential for the paper. A brief description of the specificity would be enough.

Line 315 – This needs to be very clearly delineated as swine serum?  This may be the strongest validation of this approach in the paper.

Line 318 – What is a “level”? Do you mean titer, antibody activity, etc.?

General Comments:

The paper is well written and with attention to some details as mentioned can be a very good manuscript. However, it lacks one fundamentally important bit of data: Are these epitopes immunogenic in swine and would or do they generate a potentially protective response in swine. ASFV is a porcine pathogen, and all immunological data needs to be validated in context of host species responses.

Author Response

Responses to reviewers

We would like to thank the reviewers for their valuable comments that greatly improved our manuscript. We revised the manuscript and figures according to reviewers’ comments, and all revisions in the manuscript are highlighted in yellow. For your convenience, reviewers' comments are in bold case, and our responses are in Blue. Please find below the response to your specific comments:

Review Report (Reviewer 1)

Comments and Suggestions for Authors

Line 47 – sentence structure is confusing. Subject verb agreement is incorrect. This sentence needs to be totally rewritten.

Answer: We sincerely apologize for  this mistake. We have made revisions in the manuscript (Pages 2, lines 47).

Line 67 – “elusive” is not an appropriate word choice. Suggestion: the precise role of 66 CD2v-IR in ASFV infection has not been determined.

Answer: Thanks for your suggestion. We have made revisions in the manuscript (Pages 2, lines 68-69).

Line 148 – Why were these specific peptides selected? The logic for selecting the primers for peptide synthesis needs to be explained further and justified. This selection process is key to the work.

Answer: We added illustration of the details in the revised manuscript as suggested (Pages 3-4, lines 142-153).

Line 150 – What proteins and why do you need the Western blots? This will improve readability.

Answer: We added illustration of the details in the revised manuscript as suggested (Pages 4, lines 159-165).

Line 202 – a flow chart of peptide vaccine production would be useful. This is actually a relatively complex approach to vaccine production – even for experimental purposes.

Answer: We have added a flow diagram that peptides conjugated BSA in the supplementary figures(Figure S1).

Line 215 – please explain this phrase more clearly: and nonimmune as a blank control 215 group.

Answer: We have made revisions in the manuscript (Pages 6, lines 228-230).

Line 220 – figure #’s are out of order – this is confusing.

Answer: We have adjusted the order of the figures and made revisions in the manuscript.

Line 224 – reference?

Answer: We have inserted references in the revised manuscript (Pages 6, lines 240-241).

Line 247 – Justify the statistical analysis approach. Are the data expected to be normally distributed, were they so distributed?

Answer: We have corrected in the manuscript. Statistical analyses were performed using unpaired t-tests or two-way analysis of variance (ANOVA) F-statistics. The data were expected to be normally distributed (Pages 7, lines 265-267).

Line 282 – I am not sure the description of the technical work describing generation of the mAb is essential for the paper. A brief description of the specificity would be enough.

Answer: Thanks for your concern. The description of the monoclonal antibody preparation test is for B cell epitope recognition and screening.

Line 315 – This needs to be very clearly delineated as swine serum? This may be the strongest validation of this approach in the paper.

Answer: Thanks for your concern. This needs to be delineated as swine serum, which we purchased from China Veterinary Drug Supervision Institute.

Line 318 – What is a “level”? Do you mean titer, antibody activity, etc.?

Answer: Thanks for your concern. We amended the title in the manuscript as “The epitope 264EPSPREP270 induced high antibody titer in humoral immune response” (Pages 11, lines 340).

General Comments:

The paper is well written and with attention to some details as mentioned can be a very good manuscript. However, it lacks one fundamentally important bit of data: Are these epitopes immunogenic in swine and would or do they generate a potentially protective response in swine. ASFV is a porcine pathogen, and all immunological data needs to be validated in context of host species responses.

Answer: We agree that it is important to evaluate immunological responses in swine in vivo. Since our institute does not have high-level biosafety laboratory for manipulating live ASFV. We first used mouse as a small animal model evaluating the immune effect of the epitope, which provides material basis for the next investigation in the future.

Reviewer 2 Report

The manuscript submitted by Lu et al. entitled "A Novel Epitope Located at the Intracellular Region of ASFV CD2v Elicits Humoral and Cellular Immune Responses" aims to identify and validate a novel antigenic etitope at the ASFV-CD2v protein. The manuscript is well-written and the results are well-presented and discussed. However, in the opinion of this reviewer, the following points should be improved before the acceptance of the manuscript:

Line 54 - the recent review about the current situation of ASFV vaccines should be added (10.1080/22221751.2022.2108342);

Figure 1 - Please comment on the upper bands observed in D and E

Figure 2A - Elaborate on the unspecific upper bands 

Figure 2D - Add DAPI staining to understand if CD2v colocalizes in the nucleus.

Figure 4 B and D - please keep the order between A and B and C and D. Addicionally, correct the name of the peptide 264EPSPREP270 

English is OK

Author Response

Responses to reviewers

We would like to thank the reviewers for their valuable comments that greatly improved our manuscript. We revised the manuscript and figures according to reviewers’ comments, and all revisions in the manuscript are highlighted in yellow. For your convenience, reviewers' comments are in bold case, and our responses are in Blue. Please find below the response to your specific comments:

Review Report (Reviewer 2)

Comments and Suggestions for Authors

The manuscript submitted by Lu et al. entitled "A Novel Epitope Located at the Intracellular Region of ASFV CD2v Elicits Humoral and Cellular Immune Responses" aims to identify and validate a novel antigenic etitope at the ASFV-CD2v protein. The manuscript is well-written and the results are well-presented and discussed. However, in the opinion of this reviewer, the following points should be improved before the acceptance of the manuscript:

Line 54 - the recent review about the current situation of ASFV vaccines should be added (10.1080/22221751.2022.2108342);

Answer: Thanks for your suggestion. We have made revisions in the manuscript (Pages 2, lines 52-55).

Figure 1 - Please comment on the upper bands observed in D and E

Answer: We believe that CD2v-IR protein dimerization occurs during the purification process. We partially eliminated some of the dimerization by using gel filtration chromatography (Pages 7, lines 277-279).

Figure 2A - Elaborate on the unspecific upper bands

Answer: We speculated that there were more miscellaneous proteins in the samples after cell lysis, which may be caused by non-specific reaction between non-target proteins and antibodies (Pages 8, lines 294-296).

Figure 2D - Add DAPI staining to understand if CD2v colocalizes in the nucleus.

Answer: We have added the DAPI staining in Figure 3E and made revisions in the manuscript (Pages 5, lines 184-187; Pages 8, lines 298-301), which is consistent with previous studies (Jiang et al., 2023, Viruses, 15(1):131. doi: 10.3390/v15010131). 

Figure 4 B and D - please keep the order between A and B and C and D. Addicionally, correct the name of the peptide 264EPSPREP270

Answer: We have made revisions in the Figure 5.

Round 2

Reviewer 1 Report

Really no further technical comments, the additions are good and help clarify many points. There is still a need to edit word choice and sentence structures.

Some issues with word choices, but this can be revised subject to the editors discretion.

Author Response

Responses to editors and referees

We would like to thank the editors and reviewers for their valuable comments that greatly improved our manuscript. We revised the manuscript and figures according to comments, and all revisions in the manuscript are highlighted in yellow. For your convenience, editors’ and reviewers' comments are in bold case, and our responses are in blue. Please find below the response to your specific comments:

Academic Editor Comments for Author

Point 1: The main figures are not available on the website. Could you upload the figure file?

Response: Thanks for your suggestion. We will upload the main figures as suggested.

Point 2: The format is not acceptable for review. Could you review the submission guidelines?

Response: Thanks for your suggestion. We have modified the format of all figures in the revised manuscript. We have added “Ethics Statements” to the “Materials and methods” in the revised manuscript (Page 2, lines 79-82, highlighted in the revised manuscript). We added “Institutional Review Board Statementat” and “Informed Consent Statement” in the revised manuscript (Page 14, lines 446-463, highlighted in the revised manuscript).

Review Report (Reviewer 1)

Comments and Suggestions for Authors

Point 1: Really no further technical comments, the additions are good and help clarify many points. There is still a need to edit word choice and sentence structures.

Response: Thanks for your suggestion. We have carefully reviewed the documents and addressed the language editing issues in the revised manuscript (highlighted in the revised manuscript).

Reviewer 2 Report

The authors answered to all raised questions in a positive fashion, thus the manuscript reaches a suitable form to be pusblished.

Author Response

Responses to editors and referees

We would like to thank the editors and reviewers for their valuable comments that greatly improved our manuscript. We revised the manuscript and figures according to comments, and all revisions in the manuscript are highlighted in yellow. For your convenience, editors’ and reviewers' comments are in bold case, and our responses are in blue. Please find below the response to your specific comments:

Academic Editor Comments for Author

Point 1: The main figures are not available on the website. Could you upload the figure file?

Response: Thanks for your suggestion. We will upload the main figures as suggested.

Point 2: The format is not acceptable for review. Could you review the submission guidelines?

Response: Thanks for your suggestion. We have modified the format of all figures in the revised manuscript. We have added “Ethics Statements” to the “Materials and methods” in the revised manuscript (Page 2, lines 79-82, highlighted in the revised manuscript). We added “Institutional Review Board Statementat” and “Informed Consent Statement” in the revised manuscript (Page 14, lines 446-463, highlighted in the revised manuscript).

Review Report (Reviewer 2)

Comments and Suggestions for Authors

Point 1: The authors answered to all raised questions in a positive fashion, thus the manuscript reaches a suitable form to be pusblished.

Response: Thanks for your comment.
